# Improvement of Gene Delivery and Mutation Efficiency in the CRISPR-Cas9 Wheat (*Triticum aestivum* L.) Genomics System via Biolistics

**DOI:** 10.3390/genes13071180

**Published:** 2022-06-30

**Authors:** Jaclyn Tanaka, Bastian Minkenberg, Snigdha Poddar, Brian Staskawicz, Myeong-Je Cho

**Affiliations:** 1Innovative Genomics Institute, University of California, Berkeley, CA 94704, USA; jtanaka89@gmail.com (J.T.); bminkenberg@gmail.com (B.M.); snigdhapoddar@gmail.com (S.P.); stask@berkeley.edu (B.S.); 2Department of Molecular and Cell Biology, University of California, Berkeley, CA 94720, USA; 3Department of Plant and Microbial Biology, University of California, Berkeley, CA 94720, USA

**Keywords:** wheat, transformation, biolistics, microparticle size, rupture disk pressure, high temperature, genome editing, phytoene desaturase gene (PDS), albino phenotype

## Abstract

Discovery of the CRISPR-Cas9 gene editing system revolutionized the field of plant genomics. Despite advantages in the ease of designing gRNA and the low cost of the CRISPR-Cas9 system, there are still hurdles to overcome in low mutation efficiencies, specifically in hexaploid wheat. In conjunction with gene delivery and transformation frequency, the mutation efficiency bottleneck has the potential to slow down advancements in genomic editing of wheat. In this study, nine bombardment parameter combinations using three gold particle sizes and three rupture disk pressures were tested to establish optimal stable transformation frequencies in wheat. Utilizing the best transformation protocol and a knockout cassette of the phytoene desaturase gene, we subjected transformed embryos to four temperature treatments and compared mutation efficiencies. The use of 0.6 μm gold particles for bombardment increased transformation frequencies across all delivery pressures. A heat treatment of 34 °C for 24 h resulted in the highest mutation efficiency with no or minimal reduction in transformation frequency. The 34 °C treatment produced two M_0_ mutant events with albino phenotypes, requiring biallelic mutations in all three genomes of hexaploid wheat. Utilizing optimal transformation and heat treatment parameters greatly increases mutation efficiency and can help advance research efforts in wheat genomics.

## 1. Introduction

Wheat is grown on more land than any other crop and is the second most produced grain in the world behind maize. It serves as a staple food worldwide, accounting for a fifth of globally consumed calories (http://www.fao.org/faostat/en, (accessed on 2020)). It is an important source of carbohydrates and is the leading source of vegetable-based protein in the human diet. World trade for wheat is greater than all other grains combined, which, in effect, has a great impact on global food security. With looming changes to the environment brought on by climate change as well as a growing global population, the need to address issues such as drought, yield and disease in wheat is critical. Our ability to edit wheat to produce more robust plants that can take on changing environmental landscapes and societal needs is imperative.

Bread wheat is genetically hexaploid [1], which makes breeding more complicated than other cereals such as rice and maize as a result of the triple genomes. Traditionally, wheat has been bred by crossing two lines by hand and subsequently segregating progeny for the desired traits. This task is time consuming and takes multiple generations to achieve the necessary genetic composition due to the hexaploid nature of wheat. As such, it can take years for a wheat cultivar to be established for commercial use through traditional breeding.

More modern development of wheat varieties is obtained through indirect or direct gene transfer, typically via *Agrobacterium* and particle bombardment, respectively. Through *Agrobacterium*-mediated transformation, selected genes are transferred from bacteria to plant cells via disarmed *Agrobacterium* vectors [2,3], and whole plants are then generated through tissue culture. *Agrobacterium*-mediated transformation has been well established in a variety of crops and is advantageous for intact transfer of larger DNA fragments [4]. Although *Agrobacterium*-mediated transformation can result in high instances of single-copy events and intact T-DNA delivery, it lacks consistency across species, genotypes and tissue types, especially in more recalcitrant varieties [5].

Conversely, particle bombardment is a method of direct gene transfer in which DNA is precipitated onto gold particles and delivered onto plant tissue using high pressure helium gas [6,7,8,9]. In this system, there are multiple factors that can be adjusted to optimize DNA delivery such as the size of gold particles, amount of gold particles, delivery pressure, amount of DNA and distance from the plant tissue. We chose to use particle bombardment as our gene delivery system because it is less species- and genotype-dependent and vector construction is simpler. Although particle bombardment can have higher instances of multiple copy events, the copies are often located on the same locus, allowing for easy segregation in future generations. Choi et al. [10] showed that 18 out of 19 independent transgenic barley events generated via bombardment had transgene integration at a single locus. In addition, the optimized protocol can be used to improve editing efficiency in plants via bombardment of ribonucleoproteins (RNPs) for DNA-free gene editing [11].

In the forefront of genetic engineering today is gene editing and CRISPR [12]. The CRISPR-Cas9 gene editing system is widely used for the improvement of various field crops [13]. The system allows researchers to utilize short repeats of endogenous DNA in the plant genome derived from bacteriophages to identify specific locations in the genome for gene editing. In conjunction with these DNA repeats, the Cas9 protein is programmed to cut double-stranded DNA in precise locations and allows for site-specific editing within the plant genome. The ability to make site-specific gene edits in a plant gives this technology an advantage over traditional genetic modification methods that randomly insert DNA. The CRISPR-Cas9 system allows for not only the insertion of new DNA into the plants, but also the deletion or silencing of single genes within the genome that can confer a variety of advantageous phenotypes.

Mutation efficiencies using the CRISPR-Cas9 system vary widely across monocot species. Rice mutation efficiencies are generally higher compared to wheat mutation efficiencies [14,15,16,17]. In addition, mutation efficiencies are largely impacted by the individual components of a construct. The proper selection of promoters to drive expression of Cas9 and sgRNAs can increase expression levels and ultimately positively impact mutation efficiency in transformed plants. In addition, testing sgRNA sequence efficiency in vivo before stable transformation is critical for maximizing mutation efficiency [18]. Even the sequence of the Cas9 protein can affect mutation efficiency, through codon optimization and the presence or absence of introns [19]. At the plant level, mutation efficiency is dependent on establishing an effective tissue culture protocol which is predominantly reliant on identifying a good starting explant. Another approach to increase mutation efficiency in plants is the effect of temperature treatments. The effect of temperature treatments on mutation efficiencies has been proven in mammalian cell culture [20]. The CRISPR-Cas9 system was established on a principle derived from *Streptococcus pyogenes* adaptive immunity to viruses [12]. *S. pyogenes* grows the most dynamically at 40 °C [21]. It is reasonable to expect that the Cas9 protein will be more efficient at higher temperatures. Recent studies have reported positive effects of temperature treatment on editing efficiency in *Arabidopsis*, citrus, rice and wheat plants [22,23,24].

The phytoene desaturase gene (PDS) is commonly applied as a demonstration of experimental mutation efficiencies due to its visual phenotype and wide conservation across species. The PDS gene is involved in the carotenoid synthesis pathway in plants [25]. A recessive mutation, or knockout, of the PDS gene disrupts the formation of β-carotene and confers a visual albino phenotype [26]. The PDS knockout has been demonstrated in a range of species including, but not limited to, *Arabidopsis* [26], rice [27], banana [28], cassava [29] and melon [30]. However, the PDS knockout, the albino phenotype, has not been previously reported in M_0_ hexaploid wheat because it is likely to require biallelic mutations on all six loci of the three genomes.

Finding a good combination of these factors to increase CRISPR mutation efficiency in wheat is a valuable tool for facilitating the production of robust wheat cultivars that can withstand the effects of climate change faster than other approaches. In this study, we establish parameters for particle bombardment that result in improved transformation frequencies, as well as subsequent temperature treatments to increase mutation efficiencies in hexaploid wheat. We also report the successful generation of PDS triple recessive mutant events in the M_0_ generation displaying the albino phenotype. In addition, we demonstrate the albino phenotype in M_1_ and M_2_ progeny plants derived from an edited event with monoallelic and biallelic mutations in the three different genomes.

## 2. Materials and Methods

### 2.1. Plant Material

Seeds of *Triticum aestivum* L. cv. Fielder were sown weekly and grown in growth chambers under 16-h days at 24 °C, and 8-h nights at 15 °C. Light levels were set to approximately 130 μmol m^−2^ s^−1^ at head height. Immature spikes were harvested 10–14 days post flowering with an immature embryo (IE) sized 1.7–2.2 mm. Immature spikes were collected up to 5 days pre bombardment and stored at 4–6 °C. One day prior to bombardment, immature seeds were harvested from immature spikes and surface-sterilized using 20% (*v*/*v*) bleach (8.25% sodium hypochlorite) plus one drop of Tween 20 for 15 min before triple rinsing with sterile water. IEs were then isolated and placed scutellum side up on DBC3 medium [31], and incubated at 26 °C overnight.

### 2.2. Plasmids

Plasmids, pAct1IHPT-4, pAct1IDsRED and pRGE610-PDS-PS2, were used for transformation (Figure 1). pAct1IHPT-4 [31] and pAct1IDsRED contain hygromycin phosphotransferase (*hpt*) and *DsRED* genes, respectively, each under control of the rice actin 1 promoters, its intron (*act1I*) and the nos 3′ terminator (Figure 1A,B). pRGE610-PDS-PS2 contains Cas9 gene and gRNA cassettes (Figure 1C) and was made using the following steps. First, the wheat U6 promoter with blue-white screening cassette was amplified from pTaU6-sgRNA (gift from Daniel Voytas) with primers TaU6Lac-*Hin*dIII-F (5′-TAAAGGAACCAATTCAGTCGACTGGAT-3′) and TaU6Lac-*Sbf*I-R (5′-GCCCTGCAGGTCTAGATATCTCGAGGGTACCAAACTGAG-3′). pRGE32 (gift from Yinong Yang, Addgene ID 63159) was digested with *Sbf*I and *Hin*dIII to release the original sgRNA cassette. Then, the PCR fragment from the first step was ligated into the digested pRGE32 backbone to create pRGE610. Next, two fragments from pGTR (gift from Yinong Yang, Addgene ID 63143) were amplified. The first fragment, a product of TaU6-L5AD-*Btg*ZI-F (5′-CGGGTCTCACTTGGCGATGTCTTGGTCTGCTTGACAAAGCACCAGTGG-3′) and PDS-PS2-gR (5′-TAGGTCTCAAGGTGGTCATTGCACCAGCCGGG-3′), contained a tRNA and the first half of the sgRNA spacer PS2 with 4 bp overhangs on each site to enable Golden Gate assembly with the second fragment. The second fragment, a product of PDS-PS2-gF (5′-CGGGTCTCCACCTTCTTTTCAGCGTTTTAGAGCTAGAA-3′) and L3AD-*Btg*ZI-R (5′-TAGGTCTCCAAACGCGATGGAGCGACAGCAAACAAAAAAAAAAGCACCGACTCG-3′), contained the second half of PS2 followed by the sgRNA scaffold and overhangs for Golden Gate assembly on each site. The NEB^®^ Golden Gate Assembly Kit (BsaI-HF^®^v2) was used to combine both fragments. The resulting product was used as a template for primers TaU6-S5AD-*Btg*ZI-F (5′-CGGGTCTCACTTGGCGATGTCTTGGTCTGCTTG-3′) and S3AD-*Btg*ZI-R (5′-TAGGTCTCCAAACGCGATGGAGCGACAGCAAAC-3′) to yield enough of the Golden Gate assembly product for digestion with *Btg*ZI. The *Btg*ZI digested product was then ligated into the *Bsa*I digested backbone of pRGE610, to finally create pRGE610-PDS-PS2.

### 2.3. Stable Transformation via Particle Bombardment

Immature embryos, isolated from immature spikes (Figure 2A) and pre-incubated at 26 °C overnight, were used for bombardment. On the day of bombardment, IEs were placed on top of a 40 mm filter paper on a plate of DBC3 osmoticum medium containing mannitol and sorbitol (0.2 M each) (Figure 2B) [32]. Four hours after treatment with osmoticum, IEs were bombarded using a Bio-Rad PDS-1000/He particle gun (Figure 2C) as previously described [8,32], with modifications. Two milligrams of gold particles (0.4, 0.6 and 1.0 μm) were coated with 5 μg of a mixture of pAct1IHPT4 and pAct1IDsRED or pAct1IHPT4 and pRGE610-PDS-PS2 at a 1:2 ratio. Each particle prep was resuspended in 85 μL of 100% EtOH, and 7.5 μL was spread onto the center of a macrocarrier inside of a macrocarrier holder. The particle preps were used for bombardment with a Bio-Rad PDS-1000/He biolistic device (Bio-Rad, Hercules, CA, USA) at 3 different delivery pressures (650, 900 and 1100 psi). Each plate of IEs was bombarded twice per treatment. After bombardment, IEs were transferred from filter paper to the exposed media and incubated overnight at 26 °C in dim light (10–30 μmol m^−2^ s^−1^). Sixteen hours post-bombardment, IEs were transferred to DBC3 medium and incubated at 26 °C for 1 wk in dim light. Following the resting period, IEs went through 3 rounds of selection via DBC3 media containing 30 mg/L hygromycin B, each round of selection lasting 3 wk (Figure 2D). After the third round of selection, regeneration was initiated using DBC6 media [33] containing 30 mg/L hygromycin B and incubated at 26 °C in high light (90 μmol m^−2^ s^−1^), and subcultured every 3 wk. Once shoots were approximately 0.5–3.0 cm in height (Figure 2E), shoots were transferred to WR rooting media for root formation. Plantlets were then transferred to soil once they had enough shoots to support transplant to soil (Figure 2F).

### 2.4. Temperature Treatment of Bombarded Immature Embryos

Bombarded IEs were subjected to varying heat treatments of 26 °C, 30 °C, 34 °C and 37 °C, 4 days post-bombardment, for 24 h, incubated in Heratherm Compact Microbiological Incubators (Thermo Scientific, Cat# 50125590, Grand Island, NY, USA).

Non-bombarded IEs with temperature treatments were tested for tissue culture response. The IEs were isolated onto DBC3 medium and incubated for 1 day. The following day, they were placed on DBC3 osmoticum medium for 20 h to replicate the same treatment as bombarded IEs. The IEs were then transferred back to DBC3 medium for 3 days, and then split up and treated with heat treatments of 26 °C, 30 °C, 34 °C and 37 °C for 24 h. IEs were allowed to grow and proliferate into callus for 5 weeks, at which point the callus in each plate was collectively weighed for comparison.

### 2.5. Fluorescent Visualization

Fluorescent images of IEs, calli and plantlets of transgenic Fielder events were visualized with a fluorescent Leica M165 FC stereomicroscope, equipped with Leica DFC7000 T (JH Technologies, Fremont, CA, USA), using two microscopic filters, brightfield and ET DSR with 545 nm excitation and 620 nm emission. The microscope was linked to camera imaging software, Leica Application Suite version 4.9, which was used to capture the fluorescent images. Screening of fluorescent activity was measured at different magnifications.

### 2.6. Detection of Transgenes and CRISPR/Cas9 Mutations

Genomic DNA was extracted from leaf tissue following a CTAB extraction method [34]. The *hpt, dsRED* and *Cas9* transgenes were confirmed by PCR using sequence specific primers (Appendix A). Amplifications were performed in a 25-μL reaction with DreamTaq PCR Master Mix (2X) (Thermo Fisher Scientific, Grand Island, NY, USA) as described [35]. For each PCR reaction, 23 μL were loaded onto a 0.8% agarose gel for electrophoresis.

For detection of PDS mutations, a fragment within range of the desired mutation was targeted via homoallele specific primers across each genome (Appendix A) and amplified by PCR. The amplified PCR product was cut, and DNA was extracted from gel samples using the Qiagen QIAquick Gel Extraction Kit (QIAGEN, Chatsworth, CA, USA). The purified DNA samples were used for Sanger sequencing. Mutation efficiency was calculated as the number of mutated unique events per treatment divided by the total number of events regenerated from the same treatment.

### 2.7. M_1_ and M_2_ Mutation Screening

Mutation line PC14A containing monoallelic mutations in the A and D genomes and a heterozygous biallelic mutation in the B genome was chosen for next generation progeny screening. M_1_ IEs sized 2.0–3.0 mm were harvested from the M_0_ plant. IEs were surface sterilized with 20% bleach plus one drop of Tween 20 for 15 min. Sterilized IEs were triple rinsed with sterile water to remove excess bleach. IEs were then excised and placed on WR medium scutellum side down for germination. Once plantlets germinated, progeny were visually assessed for phenotype and sampled for genotyping and mutation analysis. For M_2_ mutation screening, IEs of M plants from 3 PC14A-derived lines, PC14A-13, PC14A-24 and PC14A-27, were used for PDS phenotyping and genotyping; all 3 lines had homozygous biallelic mutations on 2 genomes and a heterozygous monoallelic mutation on the remaining genome.

## 3. Results and Discussion

### 3.1. Optimization of Particle Bombardment Parameters for Plant Transformation

#### 3.1.1. Gold Particle Size and Transformation Frequency

The DsRED visual marker was used to assess the particle bombardment parameters and stable transformation frequencies. Transient DsRED expression driven by the rice actin 1 promoter and its intron was initially detected 1 day after particle bombardment of wheat (cv. Fielder) IEs, and was clear 2 to 3 days post-bombardment (Figure 3A). DsRED-expressing sectors were formed (Figure 3B), and stably transformed plantlets were generated 6–8 weeks and 10–12 weeks post-bombardment, respectively (Figure 3C,D).

Three gold particle sizes, 0.4 μm, 0.6 μm and 1.0 μm, in conjunction with three delivery pressures, 650 psi, 900 psi and 110 psi, were tested to compare their effects on stable transformation frequency at the T_0_ plant level (Table 1). The size and weight of the individual particles has an effect on its ability to physically deliver DNA to the plant cell. There were slight variations in stable transformation frequencies across the three delivery pressures for a single gold particle size (Table 1). No single delivery pressure outperformed another across all particle sizes. In general, particle size has a greater impact on stable transformation efficiency than delivery pressure. We noted a clear data trend when comparing each gold particle size as a whole in relation to its counterparts. It was observed that the 0.6 μm gold particles performed best in transformation frequency across all delivery pressures with an average frequency of 22.6% (Table 1). The 0.4 μm particles performed second best but with an average transformation frequency of 10.7%, which measured well below that of 0.6 μm. However, the 1.0 μm particles resulted in the lowest average transformation frequency of 9.0% across all delivery pressures. In our study, we used the same weight of gold particles and the same amount of DNA per prep, meaning that the larger-sized particles will have fewer particles than the smaller-sized ones. Theoretically, 1.0 μm particles have 4.5-fold and 16.7-fold fewer particles per weight than 0.6 μm and 0.4 μm particles, respectively, because gold particle volume (weight) is calculated as (4/3) πr^3^ (r = radius) (Figure 4A). Therefore, the use of 1.0 μm gold particles resulted in a lower transformation frequency, likely due to a smaller number of particles per bombardment compared to 0.6 μm particles (Table 1). In addition, the large particles can damage the cells beyond their ability to recover, and subsequently negatively affects regeneration of transgenic plants. However, our results from the 0.4 μm vs. 0.6 μm comparison showed that transformation frequency (22.6%) with 0.6 μm particles was 2.1-fold higher than that (10.7%) with 0.4 μm particles (Table 1), even though the number of 0.6 μm particles were 3.7-fold less than that of 0.4 μm particles (Figure 4). This is possibly due to the smaller amount of DNA coated onto 0.4 μm particles or the reduced capability of 0.4 μm particles to penetrate the target cells of IEs, compared to 0.6 μm particles.

The theoretical calculation of the surface area of a sphere (4πr^2^) shows us that gold particle size positively correlates to surface area (Figure 4B). As the diameter, or radius, of a gold particle increases, so does the surface area of the sphere. This means that particle size alters the DNA-holding capacity of a single particle. From 0.4 μm to 1.0 μm diameters, the surface area increases by 6.25-fold. In our study, we used the same weight of gold particles and the same amount of DNA per prep, meaning theoretically the 1.0 μm gold particles can hold 6.25-fold more DNA than the 0.4 μm particles, allowing for a higher percentage of DNA delivery upon impact with the plant cells. Our data supports this theory, indicating that co-expression efficiency increases as particle diameter increases (Appendix A). Co-expression efficiency is calculated as the number of events visually expressing dsRED over the total number of events generated using hygromycin selection. The 1.0 μm particle size had the highest co-expression efficiency of 46.9%, 0.6 μm was in the middle with 32.4%, and 0.4 μm was the lowest with a co-expression efficiency of 14.3% (Appendix A). However, when optimizing transformation, 1.0 μm was not selected as a candidate due to its low transformation frequency regardless of its high DNA delivery performance.

#### 3.1.2. Delivery Rupture Pressure and Transformation Frequency

Different species and tissue types can require different rupture pressures to optimize transformation frequency. We tested three rupture pressures, 650 psi, 900 psi and 1100 psi, for each of the three particle sizes (Table 1). For the 0.4 μm particle size, both the 650 psi and 900 psi resulted in higher transformation frequencies of 12.9% and 11.8%, respectively, compared to 1100 psi (7.7%). Rupture pressures of 650 psi, 900 psi and 1100 psi resulted in similar frequencies for the 0.6 μm particle size at 24.3%, 21.6% and 21.9%, respectively. The 1.0 μm particle size resulted in the lowest transformation frequencies of 7.9%, 9.0% and 10.3% for the 650 psi, 900 psi and 1100 psi rupture pressures. In analyzing this data, we found that a 650 psi rupture pressure was optimal for both the 0.4 μm and 0.6 μm particle sizes, while 1100 psi was optimal for the 1.0 μm particle size. However, given the low transformation frequencies, 1.0 μm at any rupture pressure is not recommended.

Initially, we hypothesized that the smaller the particle size, the higher the rupture pressure would need to be to maximize delivery of the particles to the tissue. However, we found that even using the smallest particle size, 0.4 μm, at the lowest pressure of 650 psi was optimal. The 1.0 μm particle size is larger and heavier and performs better at 1100 psi, leaving us to conclude that a higher pressure is required to deliver larger particles to the plant tissue, indicating that larger-sized gold particles may have more resistance when penetrating the plant cells.

### 3.2. Optimization of Mutation Efficiency with High Temperature Treatments in PDS Gene-Edited Wheat

Although the Cas9 protein may be most active and efficient at 40 °C, the plant cells cannot survive a prolonged exposure to such a high temperature [21]. The key to finding the optimal temperature is one that satisfies both protein and plant. We initiated heat treatment tissue culture experiments testing callus growth of 10 Fielder IEs on standard DBC3 media after DBC3 osmoticum treatment with a temperature range of 26 °C, 30 °C, 34 °C and 37 °C for 1 day to monitor tissue morphology over time. To quantify the effect of heat treatment on the callus tissue, we weighed the tissue 35 days post-isolation. Evaluation of the callus tissue weight from each treatment allowed us to quantify the effects of the heat treatment. Both the 30 °C and 34 °C heat treatments weighed similar to the control without heat treatments at 1.59 g and 1.70 g, respectively, while the control of 26 °C weighed 1.66 g (Table 2). This demonstrated that plant cells are capable of long-term normal to accelerated growth after subjection to a slightly increased temperature for a short period of time. The 37 °C plate, however, grew at a slower rate, weighing in at only 1.37 g. This indicates that higher temperatures, even for short periods of time, negatively affects tissue growth over time in addition to tissue quality. Negative effects on the tissue growth rate and tissue quality will impact transformation frequencies, and thus the mutation efficiencies of the experiments.

We designed our PDS mutation efficiency experiment to confirm the effects of heat treatment on mutation efficiency in plants on a significant scale, side by side with our dsRED + hpt transformation frequency experiment. We chose our bombardment parameters based on the data set with the most promising transient dsRED expression and transformation efficiency. In order to quantify mutation efficiency, we used 0.6 μm gold particles at two different rupture pressures, 650 psi and 1100 psi, and tested a total of four temperature treatments, 26 °C, 30 °C, 34 °C and 37 °C, for 24 h, 4 days post-bombardment (Table 3). We reported transformation frequencies for each combination of bombardment parameters, the mutation efficiency at the transgenic plant level, meaning the total number of mutation events divided by the total number of transgenic events, as well as in relation to the donor embryos in each experiment, meaning the total number of mutation events divided by the total number of embryos initially bombarded (Table 3). We expected to see higher mutation frequencies at higher temperatures because of previously reported increased Cas9 activity at higher temperatures consistent with previous studies testing 22 °C, 28 °C, 32 °C and 37 °C in rice protoplasts, maize plants and *Arabidopsis* [22,23]. In the Mazahn study [23], temperatures between 28 °C and 32 °C proved to increase the Cas9 activity in vivo. However, their mutation efficiencies were reported via production of M_1_ mutants via heat treatment of M_0_ maize plants containing Cas9 and gRNA; transgenic plantlets were not produced via transformation in rice. Similarly, in the LeBlanc study [22], *Arabidopsis* and citrus plants were treated with four 30-h exposures to 37 °C during the vegetative growth stage, as opposed to the control of 22 °C. In their experiment, loss of GFP expression in the transgenic plantlets conferred mutation. They also reported an increase in mutation frequencies in comparison to their control, recording 12% GFP expression in plantlets treated with 37 °C as compared to the control at 89% GFP expressing plantlets. Exposing transformed embryos to a heat treatment of 28.5 °C was also previously reported in wheat, resulting in increased mutation efficiencies as compared to a control temperature of 25.5 °C [24]. Heat-treated embryos were exposed to 28.5 °C for the entirety of the callus selection phase, totaling 40 days, while the controls were exposed to 28.5 °C for 12 days and subsequently grown at 25.5 °C for the remaining 28 days of the callus phase. We included 34 °C as a treatment in our experiment because it is the temperature at which harvested seed is treated pre-germination. It is the highest temperature that minimally affects tissue morphology and survival, allowing plant cells to regenerate full M_0_ plantlets (Table 2), while the Cas9 protein is able to function at a temperature more closely aligned with its bacterial origin allowing for higher protein activity. In this study, bombarded IEs were subjected to varying heat treatments of 26 °C, 30 °C, 34 °C and 37 °C 4 days post-bombardment for 24 h (Table 3). We found that regardless of bombardment parameters, a 34 °C heat treatment has the most drastic positive effect on mutation efficiency up to 3.68-fold higher than any other temperature (Table 3). With heat treatment of 34 °C for both rupture pressures, 650 psi and 1100 psi, with a particle size of 0.6 μm, the mutation efficiencies at the transgenic event level were 17.2% and 36.8%, respectively. All other heat treatments for the same bombardment parameters were comparable. For the 650 psi and 1100 psi rupture pressure, mutation efficiencies measured 6.1% and 10.0% for the 26 °C treatment, 12.0% and 18.2% for the 30 °C temperature, and finally 11.8% and 12.5% for the 37 °C temperature, respectively. There was a trend of having higher mutation efficiencies at 1100 psi than 650 psi (Table 3). As a result of this data, we can conclude that 34 °C is the optimal temperature at which both the plant and overexpressed Cas9/gRNA can operate to achieve the highest mutation efficiency. The longer incubation at 34 °C still remains to be further evaluated for tissue culture response and mutation efficiency. Similarly, 16 h of exposure of Cas9-ribonucleoprotein (RNP) bombarded IEs to a high temperature, 30 °C or 37 °C, also resulted in increased indel formation in Pi21-, Tsn1- and Snn5-targeted M_0_ plants using 5 sgRNA-Cas9 RNPs; we achieved editing rates of 11.8–50.0% with the 30 °C treatment, 15.0–40.0% with the 37 °C treatment, compared to 5.0–26.3% with the standard 26 °C incubation [36].

Through the use of heat treatments, we were able to obtain a variety of M_0_ mutant genotypes within single genomes, as well as across multiple genomes. In the M_0_ generation, 75.0% (18/24) of the mutants produced were single genome mutations, 12.5% (3/24) were two genome mutations, and the remaining 12.5% (3/24) were triple mutants (Table 4). Of the three triple-mutants, PK3A, PK6A and PC14A, the first two events contained biallelic mutations across all three genomes, resulting in the PDS knockout albino phenotype; all mutations were out of frame. (Figure 5, Table 4 and Table 5). In order to obtain the albino PDS phenotype, biallelic mutations in all three genomes or mutations on all six loci are required (Table 5). To our knowledge, this is the first report of generating wheat plants with the albino PDS phenotype at the M_0_ level. Previous studies have achieved M_0_ triple-mutant knockouts in wheat but were not able to achieve a phenotype in the M_0_ generation [37]. Abe et al. [38] also generated triple-mutation knockouts on the *TaQsd1* gene for inhibition of preharvest sprouting in the M_1_ generation by crossing a M_0_ triple-mutant consisting of two biallelic mutations and one monoallelic mutation with wild-type Fielder, and segregating them in future generations. Both of our M_0_ biallelic triple-mutants were derived from the 34 °C heat treatment, supporting the hypothesis that heat treatment of transformed materials increases the activity of the Cas9 protein/gRNA, resulting in higher mutation efficiencies. In order to demonstrate the albino phenotype in M_1_ progeny plants derived from a PDS gene-edited event, we used event PC14A, which has heterozygous monoallelic mutations on both A and D genomes and a heterozygous biallelic mutation on the B genome (Table 4). M_1_ progeny from 2 out of 28 germinated seedlings of PC14A demonstrated the albino phenotype and showed a 15:1 segregation pattern (Figure 6, Table 6). Genotyping analysis of these two albino phenotype events resulted in homozygous biallelic mutations on both A and D genomes and a heterozygous biallelic mutation on B genome (Table 6), confirming that the albino phenotype requires biallelic mutations across all three genomes. In addition, M_2_ progeny from all three M_1_ lines tested, PC14A-13, PC14A-24 and PC14A-27, with homozygous biallelic mutations on two genomes and a heterozygous monoallelic mutation on the remaining one genome (Table 6), showed a 3:1 segregation ratio of green and albino PDS phenotype (Table 7). All M_2_ progeny plants showing the albino PDS phenotype had homozygous biallelic mutations on all three genomes (Table 7).

## 4. Conclusions

We improved transformation efficiencies in Fielder across all tested delivery pressures using 0.6 μm gold particles for bombardment. We successfully demonstrated an increase in mutation efficiency using heat treatments post-bombardment. A heat treatment of 34 °C for 24 h post-bombardment resulted in the highest mutation frequency and derived an albino PDS phenotype in the M_0_ generation of two mutant events, which requires biallelic mutations in all three genomes of hexaploid wheat. Utilizing optimal transformation parameters and a 34 °C heat treatment greatly increases mutation efficiency in hexaploid wheat and can help advance research efforts in wheat genomics. The results in this study can be applied to optimize the transformation frequency and improve mutation efficiency in other crop species.

## Figures and Tables

**Figure 1 genes-13-01180-f001:**
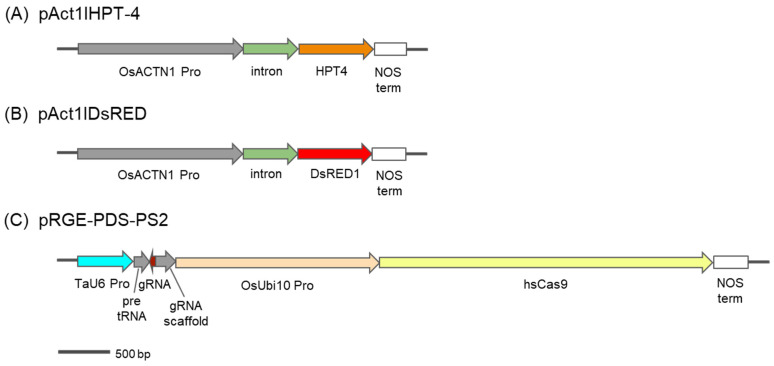
Schematic diagrams of three transformation vectors used for wheat transformation and gene editing. (**A**) pAct1IHPT-4 is a 5870-bp plasmid containing hygromycin phosphotransferase (HPT) driven by the OsAct1 promoter and its intron. (**B**) pAct1IDsRED is a 5139-bp plasmid containing DsRED driven by the OsAct1 promoter and its intron. (**C**) pRGE-PDS-PS2 is a 10,122-bp plasmid containing 2 gene cassettes where gRNA is driven by the Tau6 promoter and Cas9 is driven by the OsUbi10 promoter.

**Figure 2 genes-13-01180-f002:**
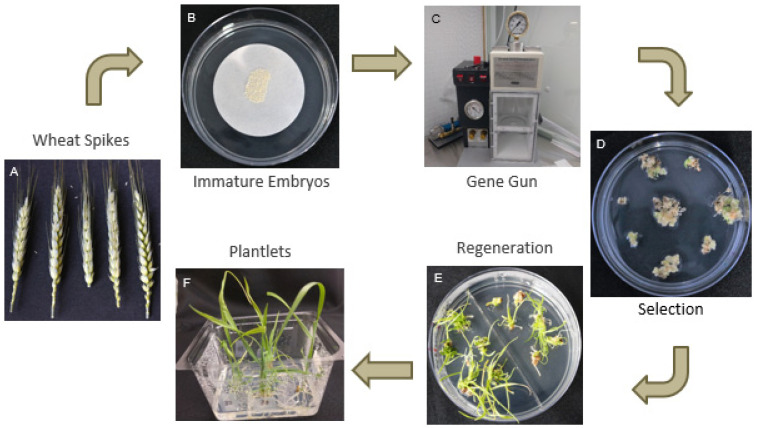
Stable wheat transformation via biolistics: (**A**) Harvest immature wheat spikes 10–14 days post-anthesis; (**B**) isolate immature embryos sized 1.7–2.2 mm and place on osmoticum medium for 4 h; (**C**) shoot gold particles using gene gun with desired gold particle size and rupture disc pressure; (**D**) plant tissue is subjected to three rounds of callus induction media containing selection, subculturing every 3 weeks; (**E**) larger callus pieces derived from a single immature embryos are broken up and placed on regeneration medium for shoot formation; (**F**) plantlets that are at least 1 cm in height are transferred to rooting medium in Phytatrays and grown to size until they can be transferred to soil.

**Figure 3 genes-13-01180-f003:**
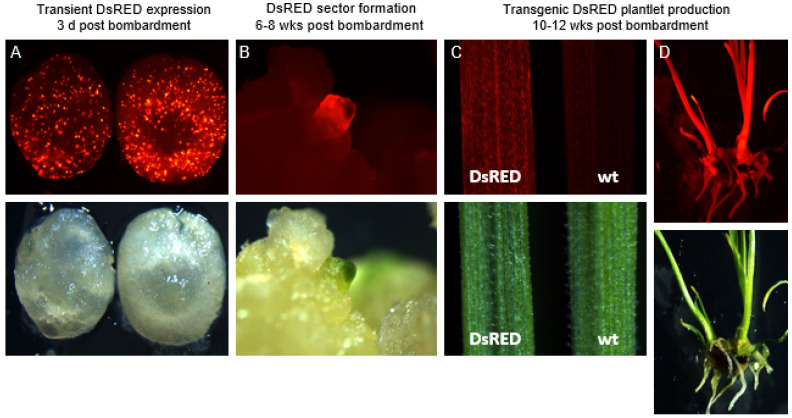
DsRED expression in different tissue types in transgenic wheat. (**A**) Transient DsRED expression and brightfield of an immature wheat embryo 3 days post-bombardment. (**B**) The formation of a DsRED sector growing 6–8 weeks post-bombardment compared with brightfield image of the same sector. (**C**) A stable transformation showing DsRED expression in leaf tissue in comparison to a wild-type control. (**D**) A stably transformed DsRED plantlet 10–12 weeks post-bombardment under fluorescence and brightfield.

**Figure 4 genes-13-01180-f004:**
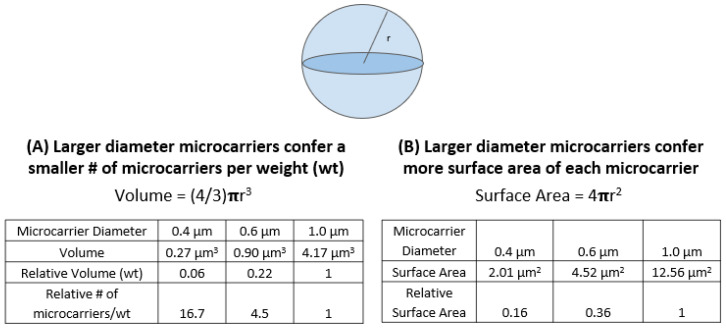
Volume (weight) and surface area of different gold particle sizes. Different gold particles are capable of holding different quantities of DNA. (**A**) The diameter of a sphere directly affects the number of gold particles by weight. As diameter increases, the number of gold particles in a fixed weight decreases. The smaller the gold particle, the more particles will be available to be coated in DNA for each bombardment prep. (**B**) Larger diameter directly affects surface area of each particle. The larger the diameter, the greater the surface area. The difference in diameter between 0.4 μm to 1.0 μm results in a 6.25-fold increase in surface area. This means that larger particles are capable of holding a greater amount of DNA.

**Figure 5 genes-13-01180-f005:**
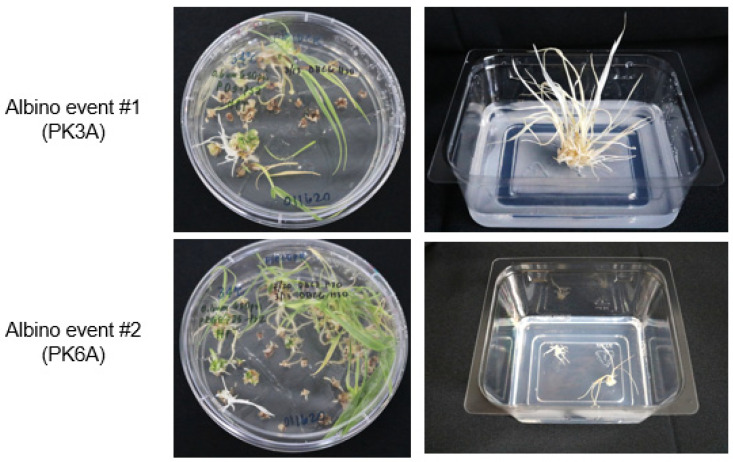
Triple biallelic knockout mutants showing albino phenotype. Photos depicting albino PDS triple biallelic mutants in plates, as well as in Phytatrays. Both albino events #1 and #2 produced as a result of the 34 °C 1-day heat treatment, shown in plate and Phytatray.

**Figure 6 genes-13-01180-f006:**
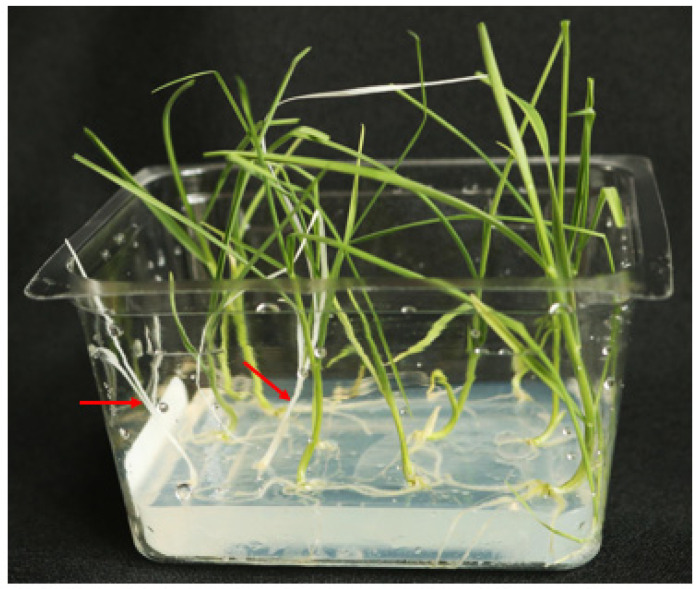
Phenotype of M_1_ progeny plants derived from event PC14A with monoallelic and biallelic mutations in the three different genomes. M_1_ progeny segregation from PC14A which contained two monoallelic mutations and one biallelic mutation. Photo of two albino M_1_ progeny of the twenty-eight total plantlets (red arrows).

**Table 1 genes-13-01180-t001:** Stable transformation frequency of wheat cv. Fielder at T_0_ plant level using different bombardment parameters. Summary of transformation frequencies with three gold particle size and three bombardment rupture pressure treatments.

	DsRED + HPT																	
	0.4 μm ^a^						0.6 μm ^b^						1.0 μm ^c^					
	650 psi		900 psi		1100 psi		650 psi		900 psi		1100 psi		650 psi		900 psi		1100 psi	
Transformation frequency *	12.9%	13/101	11.8%	13/110	7.7%	9/117	24.3%	25/103	21.6%	24/111	21.9%	25/114	7.9%	10/126	9.0%	10/111	10.3%	12/117
	10.7% (35/328)						22.6% (74/328)						9.0% (32/354)					

* Transformation frequency: (# events/# IEs bombarded) × 100%. ^a^ Column 1 shows the transformation frequencies for 0.4 μm gold particles at rupture pressures of 650 psi, 900 psi and 1100 psi, as well as the total transformation frequency of 0.4 μm as a whole. ^b^ Column 2 shows the transformation frequencies for 0.6 μm gold particles at rupture pressures of 650 psi, 900 psi and 1100 psi, as well as the total transformation frequency of 0.6 μm as a whole. ^c^ Column 3 shows the transformation frequencies for 1.0 μm gold particles at rupture pressures of 650 psi, 900 psi and 1100 psi, as well as the total transformation frequency of 1.0 μm as a whole.

**Table 2 genes-13-01180-t002:** Effect of temperature treatment on callus tissue quality and growth in wheat. Tissue culture heat treatment data. Table expressing the results of a 1-day osmoticum treatment followed by a 1-day temperature treatment of 10 embryos each, and the final weight of the tissue after 35 days of callus induction expressed in grams.

Temperature Treatment	Tissue Quality	Tissue Growth (g)
26 °C	+++++	1.66
30 °C	++++	1.59
34 °C	+++(+)	1.70
37 °C	+	1.37

‘+’ is the relative evaluation standard to quantify the regenerability of callus. +, lowest; +++++, highest.

**Table 3 genes-13-01180-t003:** Mutation efficiency with four different temperature and two bombardment rupture pressure treatments in wheat. Summary of experimental factors with associated mutation efficiencies and transformation frequencies for each data set.

Temperature Treatments ^a^	Bombardment Parameters	# Embryos Bombarded	# Transgenic Events	Transformation Frequency at T_0_ Plant Level ^b^	# Mutation Events	Mutation Efficiency at Transgenic Event Level ^c^	Mutation Efficiency at Donor Embryo Level ^d^
26 °C	0.6 μm 650 psi	219	33	15.1%	2	6.1%	0.9%
	0.6 μm 1100 psi	130	20	15.4%	2	10.0%	1.5%
	**Subtotal**	**349**	**53**	**15.2%**	**4**	**7.5%**	**1.1%**
30 °C	0.6 μm 650 psi	220	25	11.4%	3	12.0%	1.4%
	0.6 μm 1100 psi	131	11	8.4%	2	18.2%	1.5%
	**Subtotal**	**351**	**36**	**10.3%**	**5**	**13.9%**	**1.4%**
34 °C	0.6 μm 650 psi	221	29	13.1%	5	17.2%	2.3%
	0.6 μm 1100 psi	131	19	14.5%	7	36.8%	5.3%
	**Subtotal**	**352**	**48**	**13.6%**	**12**	**25.0%**	**3.4%**
37 °C	0.6 μm 650 psi	223	17	7.6%	2	11.8%	0.9%
	0.6 μm 1100 psi	131	8	6.1%	1	12.5%	0.8%
	**Subtotal**	**354**	**25**	**7.1%**	**3**	**12.0%**	**0.8%**

^a^ Temperature treatment—1 day temperature treatment at stated temperature, 4 days post-bombardment. ^b^ Transformation frequency at T_0_ plant level—calculated as the number of regenerable transgenic events produced divided by the total number of embryos bombarded. ^c^ Mutation efficiency at transgenic event level—calculated as the total number of mutation events divided by the total number of regenerable transgenic events. ^d^ Mutation efficiency at donor embryo level—calculated as the total number of mutation events divided by the total number of embryos bombarded.

**Table 4 genes-13-01180-t004:** Mutation patterns and phenotypes in genome-edited wheat events generated by four different temperature and two bombardment rupture pressure treatments.

			Mutation			
Temperature Treatments	Bombardment Parameters	Event Name	A Genome	B Genome	D Genome	Phenotype
26 °C	0.6 μm 650 psi	PI9A	WT	WT	^a^ 1-bp in	Green
		PI22A	WT	^a^ 1-bp del	WT	Green
	0.6 μm 1100 psi	PA1C	WT	Mutation	WT	Green
		PA11A	^a^ 1-bp del	WT	WT	Green
30 °C	0.6 μm 650 psi	PJ3A	WT	^c^ 5-bp del	WT	Green
		PJ7B	WT	WT	^a^ 1-bp in	Green
		PJ33A	WT	^a^ 1-bp in (1-bp rep)	WT	Green
	0.6 μm 1100 psi	PB9C	WT	WT	^a^ 1-bp in	Green
		PB10A	WT	WT	mutation	Green
34 °C	0.6 μm 650 psi	PK3A	^c^ 2-bp del	^c^ 19-bp del	^b^ 4-/4-bp del	Albino
		PK5A	^a^ 17-bp del	^b^ 3-/8-bp del	WT	Green
		PK6A	^b^ 1-bp in/13-bp del	^b^ 5-bp del (1-bp rep) /5-bp del	^b^ 1-bp in/4-bp del	Albino
		PK7A	WT	WT	^c^ 3-bp del	Green
		PK20B	Mutation	WT	WT	Green
	0.6 μm 1100 psi	PC4C	WT	WT	^a^ 1-bp in (1-bp rep)	Green
		PC5A	WT	^a^ 13-bp del	^b^ 2-bp in/1-bp del	Green
		PC6A	^a^ 5-bp del	Mutation	WT	Green
		PC6B	WT	Mutation	WT	Green
		PC8C	WT	^a^ 2-bp in	WT	Green
		PC11A	WT	WT	^a^ 1-bp in	Green
		PC14A	^a^ 8-bp del	^b^ 1-/4-bp del	^a^ 1-bp in	Green
37 °C	0.6 μm 650 psi	PL1B	WT	^b^ 3-bp del/8-bp del	WT	Green
		PL6A	WT	^a^ 3-bp del	WT	Green
	0.6 μm 1100 psi	PD5A	WT	^a^ 1-bp in	WT	Green

WT: wild-type, ^a^ heterozygous monoallelic, ^b^ heterozygous biallelic, ^c^ homozygous biallelic.

**Table 5 genes-13-01180-t005:** Molecular analysis of M_0_ PDS knockout plantlets showing albino phenotype. Molecular analysis of two albino phenotype PDS mutation events, indicating biallelic mutations across all genomes and sequences.

Albino PDS Event	Genome	Mutation Type	Sequence	Biallelic Mutation on All 3 Genomes
PK3A	A	homozygous biallelic	ATGACCACCTTCTTT- -AGCAGGTATGTC	2-bp deletion
	B	homozygous biallelic	AT- - - - - - - - - - - - - - - - - - -GGTATGTC	19-bp deletion
	D	heterozygous biallelic	ATGACCACCTTCT- - - -AGCAGGTATGTC	4-bp deletion
			ATGACCACCTTCTTTT - - - - AGGTATGTC	4-bp deletion
PK6A	A	heterozygous biallelic	ATGACCACCTTCTTTTCAAGCAGGTATGTC	1-bp insertion
			ATGACCAC- - - - - - - - - - - - -GGTATGTC	13-bp deletion
	B	heterozygous biallelic	ATGACCACCTTT- - - - - AGCAGGTATGTC	5-bp deletion/1 bp (C->T) replacement
			ATGACCACCTTC- - - - -AGCAGGTATGTC	5-bp deletion
	D	heterozygous biallelic	ATGACCACCTTCTTTTCGAGCAGGTATGTC	1-bp insertion
			ATGACCACCTTCT- - - -AGCAGGTATGTC	4-bp deletion

Red dot line: deleted bp, Red letter: mutated bp, Green letter: PAM site.

**Table 6 genes-13-01180-t006:** Phenotype and molecular analysis of M_1_ progeny plants derived from PC14A. PC14A is an M_0_ event with heterozygous monoallelic mutations in the A and D genomes and a heterozygous biallelic mutations in the B genome.

M_1_ Progeny	A Genome	B Genome	D Genome	Phenotype
PC14A-1	WT	homo (1 bp del)	WT	green
PC14A-2	WT	hetero^a^	hetero	green
PC14A-3	homo (8 bp del)	homo (4 bp del)	hetero	green
PC14A-4	homo (8 bp del)	hetero ^a^	homo (1b in)	albino
PC14A-5	hetero	homo (1 bp del)	hetero	green
PC14A-6	hetero	homo (1 bp del)	WT	green
PC14A-7	hetero	hetero ^a^	hetero	green
PC14A-8	homo (8 bp del)	hetero ^a^	homo (1b in)	albino
PC14A-9	homo (8 bp del)	hetero ^a^	hetero	green
PC14A-10	hetero	homo (1 bp del)	hetero	green
PC14A-11	WT	hetero ^a^	hetero	green
PC14A-12	hetero	homo (4 bp del)	hetero	green
PC14A-13	hetero	homo (1 bp del)	homo (1b in)	green
PC14A-14	hetero	hetero ^a^	WT	green
PC14A-15	nd ^b^	homo (1 bp del)	hetero	green
PC14A-16	hetero	hetero ^a^	hetero	green
PC14A-17	homo (8 bp del)	homo (4 bp del)	WT	green
PC14A-18	WT	hetero ^a^	hetero	green
PC14A-19	homo (8 bp del)	hetero ^a^	hetero	green
PC14A-20	hetero	hetero ^a^	hetero	green
PC14A-21	WT	homo (1 bp del)	hetero	green
PC14A-22	hetero	hetero ^a^	homo (1b in)	green
PC14A-23	hetero	hetero ^a^	hetero	green
PC14A-24	hetero	homo (4 bp del)	homo (1b in)	green
PC14A-25	WT	hetero ^a^	WT	green
PC14A-26	WT	homo (4 bp del)	homo (1b in)	green
PC14A-27	homo (8 bp del)	homo (1 bp del)	hetero	green
PC14A-28	hetero	homo (4 bp del)	hetero	green

^a^ heterozygous biallelic, ^b^ not determined.

**Table 7 genes-13-01180-t007:** Phenotype and molecular analysis of M_2_ progeny plants derived from three M_1_ PC14A lines. PC14A-13 and PC14A-24 are M_1_ progeny lines derived from PC14A. Both lines have homozygous biallelic mutations on both B and D genomes and a heterozygous monoallelic mutation on the A genome. PC14A-27 is an M_1_ progeny line with homozygous biallelic mutations on both A and B genomes and a heterozygous monoallelic mutation on the D genome. (**A**) Plant phenotype of M_2_ progeny plants derived from three M_1_ PC14A lines. (**B**) Genotype of M_2_ PDS albino plants derived from three M_1_ PC14A lines.

(**A**)					
**M_1_ Progeny**		**Segregation Ratio of M_2_ Progeny Plants (Green:Albino)**			
PC14A-13		13:2 *			
PC14A-24		11:4 *			
PC14A-27		11:5 *			
(**B**)					
**M_1_ Progeny**	**A Genome**		**B Genome**	**D Genome**	**Phenotype**
PC14A-13-1	homo (8 bp del)		homo (1 bp del)	homo (1b in)	albino
PC14A-13-2	homo (8 bp del)		homo (1 bp del)	homo (1b in)	albino
PC14A-24-1	homo (8 bp del)		homo (4 bp del)	homo (1b in)	albino
PC14A-24-2	homo (8 bp del)		homo (4 bp del)	homo (1b in)	albino
PC14A-24-3	homo (8 bp del)		homo (4 bp del)	homo (1b in)	albino
PC14A-24-4	homo (8 bp del)		homo (4 bp del)	homo (1b in)	albino
PC14A-27-1	homo (8 bp del)		homo (1 bp del)	homo (1b in)	albino
PC14A-27-2	homo (8 bp del)		homo (1 bp del)	homo (1b in)	albino
PC14A-27-3	homo (8 bp del)		homo (1 bp del)	homo (1b in)	albino
PC14A-27-4	homo (8 bp del)		homo (1 bp del)	homo (1b in)	albino
PC14A-27-5	homo (8 bp del)		homo (1 bp del)	homo (1b in)	albino

* Analyses using a χ²-test indicate that the segregation ratios of progeny plants for green vs albino phenotype were not significantly different from 3:1 (at α = 0.05).

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
