# Peer review of "Improvement of Gene Delivery and Mutation Efficiency in the CRISPR-Cas9 Wheat (Triticum aestivum L.) Genomics System via Biolistics"

_genes, 2022, doi:10.3390/genes13071180_

Round 1

Reviewer 1 Report

The manuscript “Improvement of Gene Delivery and Mutation Efficiency in the CRISPR-Cas9 2 Wheat (Triticum aestivum L.) Genomics System via Biolistics” reports a new protocol to improve mutation efficiency of CRISPR-Cas9 genomics system via Biolostics.

The results obtained are interesting to improve research in wheat genomics. The manuscript must be improved by some revisions, especially concerning tables and figures. The suggestions are given below:

Table 1 is unclear. The explanation should be reported in the text and not in the Table title.

The figure 4 is unclear. It would be better to do a table with data related to different gold particle size.

In table 2 the meaning of “+” must be specified in the capture

In the Table 3, the explanation should be reported in the text. The image is not sharp.

The table 4 is not sharp.

Author Response

Dear Reviewer,

You returned our manuscript, asking us to make revisions to address your concerns/comments. We have revised the manuscript to do so, and our responses to your reviews are as follows.

1. Table 1 is unclear. The explanation should be reported in the text and not in the Table title. Previously, only general data from Table 1 was described within the text. We’ve edited the text to include a description of Table 1, prior to the Table title, as well as describe more of the results and data contained within the table and how it as a whole relates to the results of the manuscript.

2. The Figure 4 is unclear. It would be better to do a table with data related to different gold particle size. We mistakenly added the Table 1 legend in the text again (a duplicate) just below the Figure 4 legend, so it could cause confusion. After removing it, we think it’s clear to explain what this figure means and why it is important.

3. In Table 2 the meaning of “+” must be specified in the capture. We added the meaning of “+” as follows: ‘+’ is the relative evaluation standard to quantify the regenerability of callus. +, lowest; +++++, highest

4. In the Table 3, the explanation should be reported in the text. The image is not sharp. We more clearly described Table 3 in the text. A sharper image of Table 3 has been inserted.

5. The Table 4 is not sharp. The table 4 has been replaced. Tables 5 and 6 were also replaced with sharper images.

We hope all these changes addressed your concerns.

Best Regards,

Myeong-Je

Reviewer 2 Report

The manuscript “Improvement of Gene Delivery and Mutation Efficiency in the CRISPR-Cas9 Wheat (Triticum aestivum L.) Genomics System via Biolistics” by Tanaka et al submitted for publication to Genes deals with very key topic like genome editing and optimization of gene delivery using important staple plant like wheat. 

Introduction:

The introduction is well written and point the main problems related to the topic.

Materials and methods

M&Ms are somehow well written.

Results and discussion:

Results and discussion are clearly presented.

Conclusion:

This part is well written.

Overall, I would like to congratulate the authors for the good job that they performed.

Author Response

Dear Reviewer,

Thank you for very much for reviewing our manuscript. Attached is a revised version of our manuscript following the other 2 reviewers' suggestions/comments.  Thanks again for handling our manuscript.

Best Regards,

Myeong-Je

Reviewer 3 Report

In the submitted manuscript “Improvement of Gene Delivery and Mutation Efficiency in the CRISPR-Cas9 Wheat (Triticum aestivum L.) Genomics System via Biolistics” Tanaka et al. attempted to improve the mutation efficiencies of CRISPR/Cas9 genome editing in wheat by optimized biolistics parameters with three gold particle sizes and three rupture disk pressures as well as four different temperature treatments. They found the transformation frequencies and mutation efficiency were increased using 0.6 μm gold particles for bombardment and using heat treatments of 34oC for 24 hours post bombardment. Overall, this is a good report with the experiments reasonably well arranged to convey the key messages. The conclusions are well supported by the data presented and some interesting results presented in the manuscript will be beneficial for improving mutation efficiency of CRISPR/Cas9 genome editing in other crops.

Major concern

As far as we know, the transformation efficiencies and mutation rates of genome editing depend upon many factors such as genotypes and targeted genes, the authors only used one genotype (Fielder) and one targeted gene (PDS) in this study, I wonder whether authors can provide more experimental data with different genotypes and targeted genes for extended utilization of this optimized method in the future.

Author Response

Dear Reviewer,

We share your concerns about overgeneralizing rates of expression from a single cultivar & target gene. Our study reports wheat transformation using Fielder alone. We will be testing some other wheat cultivars in the future. Regarding the effect of heat treatments on editing rates, we also tested Fielder with different gRNA sequences for 2 target genes and we got the results showing that the heat treatment improved the genome editing rates in both target genes, but we are planning to submit the data to another journal in a different way. So, I feel it is difficult to include them in this study. Instead, we’ve added the following results from our recently submitted paper to bioRxiv on RNP bombardment to a passage describing the positive effect of high temperature treatments on editing efficiency for multiple target genes in Fielder. “Similarly, 16 hours of exposure of Cas9-ribonucleoprotein (RNP) bombarded IEs to a high temperature, 30°C or 37°C, also resulted in increased indel formation in Pi21-, Tsn1-, and Snn5-targeted M0 plants using 5 sgRNA-Cas9 RNPs; we achieved editing rates of 11.8-50.0% with 30°C treatment, 15.0-40.0% with 37°C treatment, compared to 5.0-26.3% with standard 26°C incubation [36].

We hope this addressed your concerns.

Best Regards,

Myeong-Je

Round 2

Reviewer 3 Report

This revised version of the manuscript improved in terms of addressed my prior comments,